# Cationic Lignocellulose Nanofibers from Agricultural Waste as High-Performing Adsorbents for the Removal of Dissolved and Colloidal Substances

**DOI:** 10.3390/polym14050910

**Published:** 2022-02-24

**Authors:** Liangyi Yao, Xiangyuan Zou, Shuqi Zhou, Hongxiang Zhu, Guoning Chen, Shuangfei Wang, Xiuyu Liu, Yan Jiang

**Affiliations:** 1School of Resources, Environment and Materials, College of Light Industry and Food Engineering, Guangxi University, Nanning 530004, China; 2015392082@st.gxu.edu.cn (L.Y.); zouxiangyuann@163.com (X.Z.); zsq990711@163.com (S.Z.); zhx@gxu.edu.cn (H.Z.); shuangfei_wang1964@163.com (S.W.); 2School of Chemistry and Chemical Engineering, Guangxi University for Nationalities, Nanning 530006, China; chengn@bossco.cc; 3Guangxi Bossco Environmental Protection Technology Co., Ltd., Nanning 530007, China; 4Guangxi Key Laboratory of Chemistry and Engineering of Forest Products, Nanning 530006, China

**Keywords:** lignocellulose nanofibers, adsorption, deep eutectic solvents, cationization, dissolved and colloidal substances removal

## Abstract

The accumulation of dissolved and colloidal substances (DCS) in the increasingly closed paper circulating water system can seriously lower the productivity and safety of papermaking machines, and it has been a challenge to develop an adsorbent with low cost, high adsorption efficiency and large adsorption capacity for DCS removal. In this study, cationic lignocellulose nanofibers (CLCNF) were obtained by cationic modification of agricultural waste bagasse in deep eutectic solvents (DES) followed by mechanical defibrillation, and then CLCNF were employed as an adsorbent for DCS model contaminant polygalacturonic acid (PGA) removal. CLCNF was characterized by transmission electron microscopy, Fourier transform infrared, elemental analysis, X-ray diffraction, and thermogravimetric analysis. The analytical results confirmed the successful preparation of CLCNF with 4.6–7.9 nm diameters and 0.97–1.76 mmol/g quaternary ammonium groups. The effects of quaternary ammonium group contents, pH, contact time and initial concentration of PGA on the adsorption were investigated in a batch adsorption study. According to the results, the cationic modification significantly enhanced the adsorption of PGA by CLCNF and the adsorption performance increased with the increase of the quaternary ammonium group contents. The adsorption of PGA on CLCNF followed the pseudo-second-order and the fitted Langmuir isotherm model. The adsorption showed fast initial kinetics and the experimental maximum adsorption capacity was 1054 mg/g, which is much higher than PGA adsorbents previously reported in the literature. Therefore, CLCNF with high cationic group content developed in this paper is a promising adsorbent for DCS removal.

## 1. Introduction

The paper industry needs to reduce the consumption of freshwater resources due to the requirements of environmental protection. It is of great significance to increase the utilization of recycled white water and purposely convert it into a totally effluent-free papermaking process [1]. This will lead to an accumulation of pollutants in the system as the white water reuse rate increases. The accumulated pollutants in the water recycling system are called dissolved and colloidal substances (DCS) [2]. The composition of DCS, which mainly comes from pulp, filler, recycled water and the chemicals added during the papermaking process, is very complex [3]. DCS are also known as “anionic waste” because they are generally negatively charged substances in water [4]. Polygalacturonic acid (PGA) is one of the main sources of anionic waste in white water [5,6]. PGA was often selected as a model contaminant of DCS due to its relatively high content in DCS [7]. Excessive buildup of the DCS in process water stream may decrease the functioning of the paper machine and increase corrosion, foaming, pitch, precipitation, scaling, consumption of chemicals and the poor physical properties of the paper produced [2,8]. Therefore, it is necessary to remove these harmful substances or reduce their negative impact in order to obtain a completely closed loop water system.

The traditional method of controlling and reducing DCS in paper mills is to use cationic polyelectrolytes for neutralization, but this leads to a high consumption of hazardous chemicals [9]. As technology continues to improve, some new environmentally friendly methods have been invented. For example, membrane filtration treatment [10], dissolved air flotation [11], membrane reactor [12] and biological enzymes [13] have been proposed during recent years, but adaptability and costs have limited their application. Apart from these methods, adsorption methods gained increasing attention due to the advantages of simple and safe operations, low cost, no secondary pollution and the ability to treat water with high concentrations of waste. In current times, some adsorbents have been designed to reduce DCS content in white water through electrostatic interactions [1,8,14]. However, the current technology is still unsatisfactory, so it is necessary to find new adsorbents with larger capacity, higher adsorption efficiency and lower cost.

The vision has shifted to bio-based adsorbents, such as cellulosic nanomaterials, a promising class of adsorbents in the field of environmental remediation. Cellulosic nanomaterials are generally obtained from pretreated cellulose followed by nanofibrillation. So far, cellulosic nanomaterials have been used to adsorb and remove many types of water contaminants, such as natural organic matter [15], dye [16], heavy metals [17,18], fluoride [19], pharmaceutical agents [20] and viruses [21]. Cellulosic nanomaterials have proven to be an ideal sorbent for water contaminants. In contrast to other materials, they have the characteristics of high specific surface area, versatile surface chemistry, environmental inertness and renewability [22,23]. Conventional adsorbents usually have limited low adsorption efficiency and adsorption capacity due to the limited surface area or active sites for adsorption [24]. When the size of adsorbents is reduced to nanoscale, high specific surface area [25] and short intraparticle diffusion distance are expected to improve the situation. At the same time, its strong potential for surface chemical modification [26] means that a large number of active sites can be added. Cellulosic nanomaterials are renewable, widely sourced and environmentally inert biomaterials, and therefore pose little threat to the environment. However, cellulosic nanomaterials have not yet been investigated as a DCS adsorbent.

In general, cellulosic nanomaterials can reduce cost and improve performance by adjusting raw materials and pretreatment methods. In most cases, cellulosic nanomaterials used for water purification are obtained from purified cellulose sources, i.e., cellulose fibers where noncellulosic components (mainly lignin) have been removed [17]. This usually requires a complex and hazardous bleaching process. Therefore, a more beneficial way to produce water purification nanomaterials is directly from lignocellulosic raw materials without or with mild chemical treatment while achieving full lignocellulose utilization. Moreover, there is a preference for agricultural by-products rather than wood as lignocellulosic raw materials due to the lack of forest resources. Over 32 billion kilograms of high volume, low value and underutilized lignocellulosic biomaterial are produced from agricultural by-products annually, creating significant disposal problems [27]. In terms of pretreatment methods, deep eutectic solvents (DES) have been heralded as the most promising environmentally benign solvents to replace volatile organic solvents due to their almost null toxicity and total biodegradability [28]. DES are a fluid obtained by simply mixing two or three cheap and safe components with lower melting point than any of the original components [29]. Recently, deep eutectic solvents (DES) have been used as pretreatment mediums for production of functionalized lignocellulosic nanofibers [30,31].

The current study aims to use a green and simple strategy to prepare functionalized nanofiber adsorbents with high adsorption efficiency, large adsorption capacity and low cost from bagasse for the efficient removal of DCS model contaminant PGA. Concretely, cationic lignocellulose nanofibers (CLCNF) with different contents of quaternary ammonium groups have been prepared via cationic modification of bagasse in a DES followed by mechanical disintegration. DES is composed of aqueous tetraalkylammonium hydroxide and 1,3-dimethylurea, and Glycidyltrimethylammonium chloride was chosen as the cationization agent. The structure of CLCNF has been characterized using transmission electron microscopy, Fourier transform infrared, elemental analysis, X-ray diffraction, and thermogravimetric analysis. On the other hand, the effects of quaternary ammonium group contents, pH, contact time and initial concentration of PGA on the adsorption were investigated in a batch adsorption study. Moreover, the kinetics of adsorption and adsorption isotherms were performed to analyze the adsorption mechanism and predict adsorption capacity.

## 2. Materials and Methods

### 2.1. Materials

Between sixty to eighty mesh powder of sugarcane bagasse (cellulose: 43 wt%; hemicellulose: 30 wt%; lignin: 24 wt%) was collected after grinding and sieving. It was washed with water and ethanol alternately, then dried at 60 °C for 24 h before being used.

Tetraethylammonium hydroxide solution (TEAOH, 35 wt% in water) was obtained from TCI (Shanghai, China). 1,3-dimethylurea (1,3-DMU) and the cationization agent glycidyltrimethylammonium chloride (GTAC) were purchased from the Aladdin Industrial Corporation (Shanghai, China). The cationization agent was formulated into an 80 wt% solution for later use. Polygalacturonic acid (PGA) with a molecular weight between 25,000–50,000, sodium tetraborate and 3-Phenylphenol were purchased from Shanghai Macklin Biochemical Co., Ltd. (Shanghai, China). The stock solution of PGA (1 g/L) was made by dissolving PGA in pH 11 water adjusted with NaOH and then adjusted to pH 7 with HCl. 0.15% solution of 3-Phenylphenol in 0.5% NaOH and sodium tetraborate 0.0125 M in concentrated sulphuric acid were used as color rendering agents in adsorption experiments. To adjust the pH, 0.1 M HCl or NaOH was used. Deionized water was used throughout the experiments.

### 2.2. Preparation of CLCNF

#### 2.2.1. Cationization of Bagasse

The simplified reaction scheme of the cationization of bagasse is illustrated in Figure 1. Specifically, 1,3-DMU and TEAOH were mixed in a 2:1 (1,3-DMU: TEAOH) molar ratio at room temperature (24 °C) to obtain 90 g of transparent aqueous deep eutectic solvent (DES). Then, 10 g of sugarcane bagasse was added followed by the addition of GTAC solution (80 wt% in water). The dry weight of GTAC was 10 g, 20 g and 30 g, respectively, and the control group did not use the cationization agent. The reaction was mixed by mechanical stirring at room temperature for 8 h. At the end of the reaction, excessive water was added to terminate this reaction. The mixture was centrifuged for 20 min using a rotation speed of 4000 rpm. The supernatant was discarded and the sample was rediluted with water and then centrifuged again, and so on until the pH of the supernatant was neutral, indicating that the agents involved in the reaction were almost removed. These samples were collected and stored at 4 °C.

#### 2.2.2. Disintegration of Cationic Bagasse into Nanofibers

Mechanical defibrillationdecomposition of cationic bagasse was performed with a high-pressure homogenizer (AH-PILOT 2018, ATS Nano Technology Co., Ltd., Suzhou, China). The cationic bagasse was prepared as a 1 wt% suspension and subsequently decomposed in the high-pressure homogenizer at a pressure of 1000 bar for 30 min. The cationic bagasse directly entered the chamber of the high-pressure homogenizer and was easily decomposed to obtain cationic lignocellulose nanofibers (CLCNF). The nanofiber samples obtained with GTAC dosages of 10 g, 20 g and 30 g were named CLCNF-1, CLCNF-2 and CLCNF-3, respectively. The control group required pre-mechanical pulverization by shearing to enter the chamber of the high-pressure homogenizer for disintegration, otherwise it would block the narrow chamber. The nanofiber sample of the control group was named LCNF.

### 2.3. Characterizations

#### 2.3.1. Transmission Electron Microscopy (TEM)

The analysis of the size and the morphology of CLCNF and LCNF were performed by transmission electron microscopy (JEM-1400plus, JEOL, Tokyo, Japan). The samples were highly diluted to around 0.005 wt% by deionized water and then added dropwise to carbon coated copper grids. Finally, phosphotungstic acid solution (2 wt%) was applied as a negative stain before imaging. The average width of the nanofibers was measured using ImageJ software. The width of each nanofiber sample with standard errors was calculated based on over 100 individual nanofibers.

#### 2.3.2. Light Transmittance

Light transmittance of the nanofiber suspensions was measured using a UV-Vis spectrophotometer (Specord 50 Plus, Analytik Jena AG, Jena, Germany). Well-dispersed CLCNF and LCNF suspensions (0.1 wt%) were placed in a cuvette with an optical path of 10 mm and the percent transmittance in the 400–800 nm wavelength range was recorded.

#### 2.3.3. Fourier Transform Infrared (FTIR)

The chemical structure changes of pristine bagasse, CLCNF and LCNF were characterized using a Fourier transform infrared spectrometer (Tensor II, Bruker, Ettlingen, Germany). FTIR spectra was collected in the wavenumber range from 400 to 4000 cm^−1^ with a resolution of 2 cm^−1^.

#### 2.3.4. Elemental Analysis

The nitrogen content of CLCNF and LCNF was analyzed using an elemental analyser (Elemantar Vario EL cube, Elementar, Hanau, Germany). All samples were freeze-dried prior to analysis. Each of the introduced quaternary ammonium group contains one nitrogen, so the quaternary ammonium group of the nanofiber samples are directly related to the nitrogen content.

#### 2.3.5. Surface Charge Density

The surface charge densities of CLCNF and LCNF were determined using the polyelectrolyte titration method through a particle charge detector (PCD-05, BTG Mütek, Heidenheim an der Brenz, Germany). 10 mL of nanofiber suspension (0.01 wt% in water) was titrated with 0.001 N anion standard solution (Polyanetholesulfonic acid sodium) and 0.001 N cation standard solution (Polydadmac), and the surface charge density was calculated based on the consumption of standard solution.

#### 2.3.6. Zeta-Potential

The zeta-potential of CLCNF and LCNF was determined using a Zetasizer NanoZS instrument (Zetasizer Nano ZS90, Malvern Instruments Limited, Worcestershire, UK). CLCNF and LCNF were prepared at the same consistency of 0.1 wt% and adjusted to different pH for zeta-potential measurements. First, the zeta-potential of all samples was measured at pH 7. Then, the zeta-potential of CLCNF-3 and LCNF in the pH range of 3–11 (i.e., pH = 3, 5, 7, 9 and 11) was reported.

#### 2.3.7. Nanofibers Yield

The nanofiber yield of CLCNF and LCNF was determined and calculated using centrifugation. Briefly, the nanofiber suspension with a concentration of 0.2 wt% was centrifuged at a rate of 4000 rpm for 20 min to separate nanofibers (in supernatant) and non-nanofiber parts (in sediment). After carefully discarding the supernatant, the sediment was dried at 104 °C. A nanofiber yield of CLCNF and LCNF was calculated according to the following formula:(1)$$Nanofibers\;yield=\frac{W_{L}-W_{s}}{W_{L}}\times 100\%\;$$
where *W_L_* is the weight of dried lignocellulose in the suspension before centrifugation and *W_S_* is the weight of dried sediment after centrifugation.

#### 2.3.8. X-ray Diffraction (XRD)

The crystal structures of pristine bagasse, CLCNF and LCNF were investigated using an X-ray diffractometer (D8 Discover, Bruker AXS GmbH, Karlsruhe, Germany). Cu Kα radiation was generated at 40 kV and 30 mA. The scanning range is 5–50°, and the scanning speed is 5°/min. The crystallinity index (*CrI*) was estimated according to the patterns using the following equation [32]:(2)$$CrI=\frac{I_{200}-I_{am}}{I_{200}}\times 100\%$$
where *I*_200_ was the maximum intensity of the peak at 2θ between 22° and 23°, and *I_am_* was the minimum intensity of the amorphous cellulose at 2θ between 18° and 19°.

#### 2.3.9. Thermogravimetric Analysis (TGA)

Thermogravimetric analysis of pristine bagasse, CLCNF and LCNF was performed using a TGA-DSC/DTA analyzer (STA 449 F5, NETZSCH-Gerätebau GmbH, Selb, Germany) under a nitrogen atmosphere at a constant rate of 30 mL/min. Approximately 5 mg of dry sample was placed in an aluminum oxide pan and thermally degraded by heating from 30 °C to 850 °C at a rate of 10 °C/min.

#### 2.3.10. Viscosity

The viscosity of CLCNF-3 suspension (0.5 wt%) was measured using the rotational rheometer (Haake MARS 4, Thermo Fisher Scientific, Waltham, MA, USA). The measurements were conducted at 25 °C and at a shear rate of 0.01–1000 s^−1^.

### 2.4. Adsorption Studies

The adsorption experiments were performed in batch mode, and the performance of the nanofiber adsorbents were researched in terms of the different quaternary ammonium groups’ content of nanofibers (LCNF, CLCNF-1, CLCNF-2, and CLCNF-3), pH (3–11), initial concentration of PGA (400–800 mg/L), and adsorption time (0–6 h).

In batch adsorption experiments, 5 g of nanofiber suspensions at 0.5 wt%was mixed with PGA stock solution (1 g/L) and the total volume was adjusted as 50 mL with deionized water. Then the suspensions were mixed at 300 rpm by a magnetic stirrer. In adsorption experiments of various variables, the adsorption time was 6 h excluding the time series, and the initial concentration of PGA was 400 mg/L excluding the initial concentration series, and the pH of the solution was adjusted to 7 excluding the pH series. Next, the supernatant obtained after centrifuging the mixture was filtered using a 0.22 μm membrane syringe filter. The change in the concentration of PGA in the solution before and after adsorption was analyzed by a UV-Vis spectrophotometer (Specord 50 Plus, Analytik Jena AG, Jena, Germany). Briefly, in a similar way to a previously published method [33], 1 mL of PGA-containing sample and 5 mL of the sulphuric/tetraborate solution were mixed in a water-ice bath. The solution was reacted in a boiling water bath for 8 min, immediately cooled in a water-ice bath, and 0.1 mL of 3-Phenylphenol solution was added and turned to pink after mixing. The absorbance of the solution was measured in 524 nm by the UV-Vis spectrophotometer, and the PGA concentration was determined by comparison to a stable calibration curve (R^2^ = 0.9993).

## 3. Results and Discussion

In this study, sugarcane bagasse was directly cationized using a deep eutectic solvent (DES) as a reaction medium. Afterwards, three cationic lignocellulose nanofiber (CLCNF) samples with different positive charge contents and molar quantities of quaternary ammonium groups were prepared via mechanical disintegration, respectively denoted CLCNF-N where N increases with the surface charge content (i.e., CLCNF-1, CLCNF-2 and CLCNF-3). LCNF, as the reference sample, was also prepared by mechanical disintegration with DES-based pretreatment but without cationic modification. In this reaction, Glycidyltrimethylammonium chloride (GTAC) was chosen as a cationization agent, as it grafted quaternary ammonium groups to bagasse fibers and promoted mechanical decomposition. DES was composed of aqueous tetraalkylammonium hydroxide and 1,3-dimethylurea (1,3-DMU), providing an alkaline condition to allow cationic modification. This DES has been proven to be a green solvent because of the low toxicity and biodegradability of its components [34]. With the applied cationization method (Figure 1), the hydroxyl groups of cellulose, lignin and hemicellulose are deprotonated under the alkaline conditions provided by DES, and the active hydroxyl groups react with GTAC to generate quaternary ammonium groups. DES acted as a swelling agent, which is confirmed in the optical microscope images of bagasse and precursors of CLCNF and LCNF (no mechanical disintegration) in Appendix A. Robust lignocellulosic structures swelled and dissociated after DES-based treatment, and this dissociation phenomenon was significantly enhanced after GTAC usage increased.

### 3.1. Adsorbent Characterization

TEM images confirmed the nanoscaled structure of all the prepared nanofibril samples. Specifically, LCNF has an average width of 25.3 ± 6.7 nm (Figure 1A). Compared with LCNF, the CLCNF samples have well-individualized structures with a homogeneous size distribution (Figure 1B–D). This was more directly reflected in the width of CLCNF samples. As shown in Figure 1E, the average width for CLCNF-1, CLCNF-2 and CLCNF-3 was 7.9 ± 1.7, 5.5 ± 1.0 and 4.6 ± 0.8 nm, respectively. These results suggest that the cationization of bagasse could facilitate the mechanical disintegration process, thus resulting in nanofibers with small and homogeneous widths. Such a positive effect for mechanical disintegration is especially remarkable when upgrading the amount of GTAC in DES. Additionally, the precursor of LCNF needs to be mechanically pretreated for avoiding the blockage of the chamber for the high-pressure homogenizer. However, this is not the case with the cationized bagasse fibers, which can readily pass through the chamber of the high-pressure homogenizer without the occurrence of clogging. It is presumed that the charged groups on the cationized bagasse fibers can create electrostatic repulsion between fibirls and thus enhance the penetration of water into the fibers to create osmotic pressure. Undoubtedly, such effects would promote the mechanical defibrillation process of fibers [26,35].

The suspension transmittance is one of the main methods to indirectly evaluate the degree of fibrillation [36]. The light transmittance measurement results of the nanofiber suspensions are shown in Figure 2A. It was shown that the percentage transmittance in the UV–visible range of LCNF suspension was the lowest because of its huge fiber size. In contrast, the percentage transmittance of the CLCNF suspensions was significantly higher than the LCNF suspension and the CLCNF-3 suspension exhibited the highest transmittance. This is because the light transmittance of the suspension is related to the light-scattering phenomenon, and the light scattering is proportional to the cross section area of the particles [37]. A higher transmittance means a smaller fiber size and a higher degree of fibrillation. This result corroborates what was observed in the TEM images and it is also consistent with the visual observations of the nanofiber suspensions in Figure 2B.

In order to confirm the successful cationic modification of bagasse, a Fourier transform infrared (FTIR) experiment was carried out. Figure 3 illustrates the FTIR spectra of bagasse, LCNF and various CLCNF samples. It was shown that the new peak at 1484 cm^−1^ corresponding to the C-N stretching vibration appeared in all CLCNF samples but not in the other samples. And the epoxy ether vibrations that belong to GTAC at 1261 cm^−1^ were not present in all CLCNF samples, indicating the epoxide ring opening reaction of bagasse with GTAC [38]. This confirms the successful introduction of quaternary ammonium groups on bagasse. Another significant chemical structure change is the disappearance of the characteristic peak at 1733 cm^−1^ attributed to the C=O bond after DES treatment. This is because highly alkaline conditions of DES caused the breakage of naturally occurring ester bonds in bagasse, specifically the acetyl group of hemicellulose and the ester linkage of the carboxylic groups in the ferulic and p-coumaric acids of lignin/hemicellulose [32,39]. In addition, the broadening of the hydroxyl peak around 3300 cm^−1^ indicates that chemical modification altered the hydrogen bonding pattern of bagasse and made the bagasse more susceptible to moisture uptake from air. This will be further confirmed in the results of the thermogravimetric analysis later.

The specific number of quaternary ammonium groups grafted to bagasse was measured by elemental analysis (based on the weight percentages of N) and was also confirmed by polyelectrolyte titration. The results of elemental analysis and polyelectrolyte titration are listed in Table 1. The result shows that the quaternary ammonium group contents and surface charge densities were 0.97–1.76 mmol/g and 0.87–1.85 meq/g, respectively. The contents of the quaternary ammonium group in the same sample varied slightly due to the analyzing methods [40]. These results show that the increase of GTAC content in the reaction mixture improved the reactivity. The charged group content is a key performance indicator for nanofiber adsorbents in water purification applications, and it may directly affect the purification effect of contaminants [41]. CLCNF-3 was selected as the nanofiber with the highest cationic group content in this study. This was a fairly high value when compared with previous reports. In the literature, cationized CNF with an ammonium content of 0.134 mmol/g were obtained using 3-chloro-2-hydroxypropyl trimethylammonium chloride as the cationization agent [42]. Cationic CNF with a quaternary ammonium group content of 1.2 mmol/g were obtained from cellulose pulp etherified with a quaternary ammonium salt in water [15]. Cationic wood nanofiber with a cationic group content around 1.5 mmol/g were obtained using four different aqueous solvents containing TEAOH with different carbamides and GTAC as the cationization agent [32]. Surface quaternized cellulose nanofibrils with a trimethylammonium chloride content of 2.31 mmol/g were obtained from wood pulp [41]. In addition, LCNF were detected to contain trace amounts of N elements, which may be due to incomplete washing of TEAOH during centrifugal washing. Zeta-potential and surface charge densities both reflected the charge on the nanofiber surface. The zeta-potential and surface charge density were detected as negative for LCNF, but positive for CLCNF. A negative zeta-potential corresponds to a negatively charged surface, and vice versa. The results again confirmed that the presence of GTAC successfully introduced positively charged quaternary ammonium groups to the fibers.

In this study, nanofibers were separated and quantified by means of centrifugation. After centrifuging the nanofiber suspensions, the larger-sized fibers will sink to the bottom and the smaller-sized fibers will remain suspended in the upper layer. The nanofiber yield refers to the percentage of small size fibers with diameters in nanoscale in the total fiber weight [43]. It showed that the nanofiber yield of LCNF was extremely low at only 8.49%. The increase in the amount of GTAC caused the value of the nanofiber yield to increase linearly to at least 76.49%, with a maximum value of 92.50%. This again confirms the results observed in the TEM images.

Zeta-potential is an important and useful parameter to describe the electric potential in the solid/liquid interfacial layer of a material in aqueous solution [44]. The zeta-potential as a function of pH for LCNF and CLCNF-3 water suspensions at consistencies of 0.1 wt% was shown in Figure 4. In this way, more information on the surface charge states of representative samples LCNF and CLCNF-3 was obtained. In the studied pH range (3–11), the zeta-potential of CLCNF-3 was always positive, while LCNF was always negative. The absolute value of the zeta-potential of LCNF became larger as the pH in water increased, with values ranging from −6.49 to −19.20 mV. This was attributed to the negatively charged surface groups (i.e., hydroxyl groups) on LCNF and the interactions of protons and sodium ions with them at different pH. The zeta-potential of CLCNF-3 fluctuates in the range of 21.17 to 44.73 mV. These values were always positively attributed to the presence of quaternary ammonium functional groups on the surface of the nanofibers. The zeta-potential of CLCNF-3 was maximum at neutral pH, and a pH that is too high or too low will decrease the zeta-potential. This is similar to the phenomenon observed for another cationized cellulose nanofiber [42]. The state of the nanofiber surface charge will most likely affect its adsorption performance for anionic contaminants in water [41,45].

CLCNF-3 is predicted to be the most outstanding adsorbent due to it containing the highest surface charge density of all samples. Therefore, the rheological properties of CLCNF-3 were investigated. The rheological properties were obtained from the viscosity measurements of CLCNF-3 suspension and are shown in Figure 5. The results show that the viscosity of the CLCNF-3 suspension decreases with increasing shear rate, suggesting the typical shear thinning behavior of the nanofiber suspension [40]. This indicates that the CLCNF-3 suspension is a pseudoplastic fluid and its diffusion in water was unimpeded. This would facilitate the capture of contaminants in the wastewater by CLCNF-3. The literature reports that some samples containing the naturally hydrophobic non-derivated lignin were difficult to disperse in water [46,47]. This would have raised concerns about the diffusivity of CLCNF-3 in water affecting water-based applications but did not occur.

The crystal structure and the crystallinity index (*CrI*) of bagasse, LCNF and various CLCNF was investigated by X-ray diffraction (Figure 6). All samples exhibit a sharp high peak around 2θ = 22.2° in XRD graphs (Figure 6A), which corresponds the (0 0 2) lattice plane of cellulose I. The results indicate that all samples contain a large amount of native cellulose I, and the cellulose I crystal structure was basically retained after modification [48]. The *CrI* of various samples are shown in Figure 6B. In general, mechanical disintegration led to the destruction of the crystalline region of cellulose and the depolymerization of cellulose, resulting in a decrease in crystallinity [38]. Chemical treatment can dissolve the amorphous fraction of cellulose [49], hemicellulose [38] and lignin [50], which in turn may lead to an increase in crystallinity. The *CrI* values of bagasse and LCNF were 50.6% and 48.8%, respectively. It may be due to the combined effect of mechanical defibrillation and chemical treatment that the *CrI* of LCNF is slightly lower than that of bagasse. On the other hand, the *CrI* values of all CLCNF samples were significantly lower compared to that of bagasse, and CLCNF-3 showed the lowest *CrI* value at 26.5%. This is mainly due to the fact that the charged quaternary ammonium groups on CLCNF made a strong electrostatic repulsion between fibers making the mechanical disintegration effect amplified. Secondly, the quaternary ammonium groups on CLCNF were detected as an amorphous region that negatively affects *CrI* [40].

The TGA and DTG curves are presented in Figure 7, and they were used to assess the effect of DES treatment and cationic modification on the thermal degradation behavior. Bagasse exhibited a typical thermal degradation profile of lignocellulose. Its DTG curve shows two main peaks, a shoulder peak at 200~310 °C attributed to hemicellulose decomposition, and the most prominent peak at 310~395 °C, which is attributable to cellulose decomposition [51]. The TGA curves of the LCNF and bagasse almost overlap, indicating that they have similar thermal degradation behaviors. A different thermal degradation behavior was observed for CLCNF compared to bagasse. CLCNF lost more mass in the low temperature region from 30 to 160 °C. Bagasse lost about 8.8% of mass in water form and CLCNF lost 10.6% or more. This was attributed to the grafting of CLCNF with extremely hydrophilic quaternary ammonium groups. At the same time, the onset decomposition temperature (T_onset_) and the maximum decomposition temperature (T_max_) of CLCNF were significantly lower than that of bagasse. The T_onset_ of bagasse was 313.8 °C, and the T_onset_ of CLCNF was reduced by 6.3 to 24.5 °C in comparison. The T_max_ of bagasse was 356.1 °C, and the T_max_ of CLCNF was reduced by 20.6 to 37.8 °C in comparison. This is attributed to the fact that the quaternary ammonium groups on CLCNF contain a volatile component (NCH_2_(CH_3_)_2_), making it unstable and thus susceptible to degradation [32]. Secondly, the decrease in *CrI* also negatively affected the degradation temperature [52]. Clearly, DES treatment has little effect on the pyrolysis behavior of bagasse, but cationic modification greatly reduced its thermal stability.

### 3.2. Adsorption Studies

#### 3.2.1. Different Quaternary Ammonium Groups Content of Nanofibers Effect on PGA Adsorption

The relationship between the quaternary ammonium groups content and the ability of lignocellulose nanofiber adsorbents to adsorb PGA is presented in Figure 8. The non-modified LCNF reference displayed a very low adsorption removal rate of about 7.25%. Since PGA is a negatively charged contaminant in water, and LCNF with no quaternary ammonium group is also negatively charged in water, it is expected that the limited interaction between LCNF adsorbent and PGA would result in such a low adsorption effect. In comparison, all CLCNF adsorbents exhibited the significant ability to remove PGA by adsorption. CLCNF displayed an increased adsorption removal rate of PGA with the gradual increase of their positive charge. According to the results, the adsorption removal of PGA by CLCNF was achieved by electrostatic interactions between the positively charged quaternary ammonium group on CLCNF and the negatively charged PGA in water. This is considered reasonable based on previous studies [7]. A maximum of 96.92% of negatively charged contaminants could be adsorbed onto CLCNF-3 with the most quaternary ammonium groups. The corresponding adsorption capacity of CLCNF-3 is 775 mg/g, and this is much higher than the adsorption capacity of a polystyrene sphere (3 mg/g) [8] and a Gel-type ion exchange resin (Amberlite IRA-67) (19 mg/g) [14] as referenced in previous reports on PGA adsorbents. Therefore, CLCNF-3 was chosen as a model nanofiber adsorbent for the subsequent adsorption studies.

#### 3.2.2. pH Effect on PGA Adsorption

The pH value of the aqueous solution is usually one of the important factors affecting the adsorption effect, which usually affects the adsorption capacity by influencing the surface charge of the adsorbent. Meanwhile, the pH of real-life wastewater varies widely, so it is desired that potential adsorbent functions over a wide pH range. Figure 9 displays the percentage adsorption removals of PGA onto CLCNF-3 at different pH values. The results showed that the excellent adsorption removal rate of PGA by CLCNF-3 was in the pH range of 5–9, and the highest adsorption removal rate was 97.17% at pH 7. When the pH of the aqueous solution is low, many of the Cl^−^ that appear due to pH adjustment may be adsorbed to CLCNF-3 and occupy the adsorption sites, resulting in reduced adsorption capacity. When the pH is high, the presence of Na^+^ in the solution might lead to charge screening effects, resulting in a lower adsorption onto the CLCNF-3. Nevertheless, the removal rate of PGA still reached 76.99% even under the acidic condition of pH 3. These results show that CLCNF-3 has excellent adsorption capacity over a wide pH range, and the adsorption effect slightly depends on the pH. CLCNF-3 is better at adsorbing PGA in neutral and slightly alkaline environments, but the adsorption effect will be significantly reduced in an acidic environment.

#### 3.2.3. Adsorption Kinetics

It was perceived from the adsorption kinetics (Figure 10A) that the adsorption of PGA by CLCNF-3 was a very fast process, and the adsorption saturation was almost reached within 2 min of the start of adsorption. Such a rapid adsorption process was not unexpected, as nanofiber adsorbents usually exhibited fast adsorption responses to water contaminants [17,20]. CLCNF-3 exhibited such a strong and fast adsorption capacity because of its large surface area and abundant quaternary ammonium groups as binding sites to adsorb PGA. After the equilibration time of 180 min, the surface of the adsorbent became saturated.

By fitting pseudo-first-order (PFO) and pseudo-second-order (PSO) kinetic models to the experimental data (Figure 10B,C), this allows further description and understanding of the adsorption process. The equations of these two models are as follows [42]:

PFO:(3)$$log\left(Q_{e}-Q_{t}\right)=logQ_{e}-\frac{K_{1}}{2.303}t$$

PSO:(4)$$\frac{t}{Q_{t}}=\frac{1}{K_{2}Q_{e}^{2}}+\frac{t}{Q_{e}}$$
where *Q_t_* (mg/g) and *Q_e_* (mg/g) represent the amounts adsorbed at time *t* (min) and at equilibrium, respectively, and *K*_1_ (min^–1^) and *K*_2_ [mg/(g·min)] represents the rate coefficients for the PFO and PSO kinetic models, respectively.

The kinetic parameters calculated from the fitted equations in these figures along with the regression coefficients are listed in Table 2. The R^2^ value of the PSO rate equation (0.9999) is higher than that of the PFO rate equation (0.9727), indicating that the process of PGA adsorption by CLCNF-3 is more consistent with PSO kinetics. The excellent applicability of the PSO model to the adsorption kinetics of PGA onto CLCNF-3 implies that the rate-controlling step of adsorption is the chemisorption between adsorbent and adsorbate.

Since neither PFO nor PSO kinetic describe the process of diffusion, the Weber-Morris intraparticle diffusion model was also used to explain the adsorption process. The Weber-Morris intraparticle diffusion equation is as follows [45]:(5)$$Q_{t}=K_{i}t^{0.5}+C$$
where *K_i_* [mg/(g·min^0.5^)] is the rate constant of intraparticle diffusion; *C* is the constant related to the thickness of the boundary layer, which is in direct ratio to the effect of the boundary layer.

The fitted curve of the Weber-Morris intraparticle diffusion model is shown in Figure 10D, and the parameters are listed in Table 2. According to this model, the plots of *Q_t_* versus *t*^0.5^ must pass through the origin and yield a straight line. However, in this study, the plots of Q_t_ versus *t*^0.5^ were not linear over the whole time range, and the fit was very low. This result shows that the Weber-Morris intraparticle diffusion model was not suitable for predicting the adsorption kinetics of PGA onto CLCNF-3 over the whole range. This suggests that intraparticle diffusion is not the rate-limiting step in the adsorption process [53].

#### 3.2.4. Adsorption Isotherm

As the concentration of PGA in wastewater is often fluctuating, the adsorption capacity of the adsorbent at different concentrations was investigated. Figure 11 illustrates the adsorption of PGA as a function of initial concentration. Figure 11A shows the reduction in the percentage adsorption removals of PGA onto CLCNF-3 with increasing initial concentrations of PGA. This reduction can be attributed to the finite number of active sites available on CLCNF-3 that were occupied by adsorbed PGA, subsequently leading to a decrease in PGA adsorption. On the other hand, it was found that the maximum adsorption capacity of CLCNF-3 increased with increasing initial concentration until the initial concentration of PGA was increased to 550 mg/L. This is because a higher concentration provides a driving force to overcome all the resistances of PGA between the aqueous and solid phases, thus increasing adsorption; moreover, as the initial concentration increased, so did the number of collisions between PGA and CLCNF-3, thus improving the adsorption process.

The experimental data were fitted to the Langmuir model [54], which applies to monolayer adsorption, and the Freundlich model [55], which applies to multilayer adsorption, in order to identify the mechanisms of contaminant removal (Figure 11B–D). The respective equations for these models are:

Langmuir model:(6)$$\frac{C_{e}}{Q_{e}}=\frac{C_{e}}{Q_{m}}+\frac{1}{K_{L}Q_{m}}$$

Freundlich model:(7)$$lnQ_{e}=lnk_{F}+\frac{lnC_{e}}{n}$$
where *C_e_* and *Q_e_* are the PGA solution concentration (mg/L) and adsorption capacity (mg/g) at equilibrium, respectively, and *Q_m_* is the theoretical saturation capacity (mg/g). *K_L_* and *K_F_* are the Langmuir adsorption constant (L/g) and the Freundlich adsorption constant (L/mg), respectively, and *n* is the heterogeneity factor for the adsorption.

The fitting parameters are presented in Table 3. The R^2^ value for the Langmuir model is 0.9994 compared to only 0.5520 for the Freundlich model, demonstrating that the adsorption process is more consistent with monolayer adsorption without lateral interactions between adsorbed molecules. Moreover, the maximum experimental adsorption value (i.e., *Q_e_* = 1054 mg/g) is very close to the calculated value of *Q_m_*, the maximum theoretical adsorption value of PGA uptake for CLCNF-3.

The value for *K_L_* can be calculated from the Langmuir equation which can then be used to determine the separation factor *R_L_* using:(8)$$R_{L}=\frac{1}{1+K_{L}C_{0}}$$

The relationship between the initial concentration *C_0_* and the separation factor *R_L_* is shown in Appendix A. *R_L_* can be used to determine the favorability and feasibility of an adsorption process. An *R_L_* value between 0 and 1 is favorable for adsorption and values greater than 1 are adverse for adsorption [56]. The *R_L_* values for CLCNF-3 are all between 0 and 1 indicating that the adsorption of PGA by CLCNF-3 is a favorable process. It should be noted that as the initial concentration increased from 400 to 800 mg/L, the *R_L_* value gradually decreased from 0.0030 to 0.0015, indicating that higher concentrations promote PGA adsorption by CLCNF-3.

## 4. Conclusions

In conclusion, cationic lignocellulose nanofibers (CLCNF) with nanoscale diameters and quaternary ammonium group contents in the range of 0.97–1.76 mmol/g were successfully prepared by cationic modification of bagasse in deep eutectic solvents (DES) followed by mechanical defibrillation. Such lignocellulosic nanomaterials showed excellent adsorption performance in removing dissolved and colloidal substances (DCS) model contaminant polygalacturonic acid (PGA). The cationic modification in DES changed the surface charge of bagasse from negative to positive and obtained a large number of active adsorption sites, so CLCNF can easily adsorb the negatively charged PGA in water. Electrostatic interactions were considered as the main mechanism for capturing PGA by CLCNF in water. The kinetic process of PGA adsorption by CLCNF can be predicted by a pseudo-second-order model and the Langmuir model fitted the data well, indicating monolayer adsorption. The CLCNF adsorbent in current research has the advantages of low cost, high efficiency and large adsorption capacity, which can meet the new demand of DCS adsorbents. It is expected to be used for the adsorption of other anionic contaminants in water.

## Data Availability

Not applicable.

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
