# Peer review of "Cationic Lignocellulose Nanofibers from Agricultural Waste as High-Performing Adsorbents for the Removal of Dissolved and Colloidal Substances"

_polymers, 2022, doi:10.3390/polym14050910_

Round 1

Reviewer 1 Report

Greetings,

I have a couple of modifications that I find fit.

-Please pay attention to measuring units.

-In the Results section, page 15, you said in Figure 5 FTIR spectra can be found. I believe Figure 3 shows that. Please verify.

-Page 24 move PFO: on the next page

Author Response

Comments:

I have a couple of modifications that I find fit.

Point 1: Please pay attention to measuring units.

Response 1: Yes, we have revised accordingly.

Point 2: In the Results section, page 15, you said in Figure 5 FTIR spectra can be found. I believe Figure 3 shows that. Please verify.

Response 2: Sorry for our carelessness. FTIR spectra is Figure 3 and not Figure 5. We have revised accordingly.

Point 3: Page 24 move PFO: on the next page.

Response 3: Yes, we have revised accordingly.

Reviewer 2 Report

In this manuscript, the authors prepared cationic lignocellulose nanofibers for adsorption of PGA in water and studied their adsorption performance. Morphology, functionalities, surface charge, crystallinity of nanofibers were systematically characterized with TEM, FTIR, DLS, and XRD. Adsorption behavior of PGA on nanofibers was thoroughly studied. Therefore, I can suggest its publication in the journal after considering following comments.

  1. In UV-vis figure, the author mentioned that the weight % was same. In such case, I am wondering how the transmittance of LCNF hugely differs from that of CLCNF? Please add description regarding the absorption cross-section and references.

  1. It seems that the functionalities is important in this study. Please provide scheme of molecular structures when describing chemical modification and FTIR for the convenience of readers.
  2. The author mentioned that “ The new peak at 1484 cm-1 corresponds to the C-N stretching band, indicating that the cationization agent GTAC has grafted quaternary ammonium groups to bagass”. However, it seems like that it just confirms the existence of C-N bonding but does not confirm the “grafting”. Perhaps, the shift in the peak needs to be included before and after grafting.

  1. Typo : On page 5 in line 2 “Figure 5”->”Figure2”

Author Response

Comments:

In this manuscript, the authors prepared cationic lignocellulose nanofibers for adsorption of PGA in water and studied their adsorption performance. Morphology, functionalities, surface charge, crystallinity of nanofibers were systematically characterized with TEM, FTIR, DLS, and XRD. Adsorption behavior of PGA on nanofibers was thoroughly studied. Therefore, I can suggest its publication in the journal after considering following comments.

Point 1: In UV-vis figure, the author mentioned that the weight % was same. In such case, I am wondering how the transmittance of LCNF hugely differs from that of CLCNF? Please add description regarding the absorption cross-section and references.

Response 1: Light scattering can be caused by the nanoparticles in suspension with diameters larger than light wavelength. Therefore, CLCNF suspension exhibited significantly higher transmittance than that of LCNF suspension, ascribed from the remarkably smaller diamters of CLCNF in comparison to LCNF (Fig. 1). Moreover, description regarding to this issue has been added in the revised manuscript.

Point 2: It seems that the functionalities is important in this study. Please provide scheme of molecular structures when describing chemical modification and FTIR for the convenience of readers.

Response 2: Thanks for your suggestion. The functionalities is really important in this study. So I followed your advice and added the reaction formula of cationic modification of lignocellulose in “2.2.1 Cationization of bagasse” of Materials and Methods.

Point 3: The author mentioned that “The new peak at 1484 cm-1 corresponds to the C-N stretching band, indicating that the cationization agent GTAC has grafted quaternary ammonium groups to bagass”. However, it seems like that it just confirms the existence of C-N bonding but does not confirm the “grafting”. Perhaps, the shift in the peak needs to be included before and after grafting.

Response 3: Many thanks for this suggestion. We have revised the description regarding to this issue in the revised manuscript.

Point 4: Typo: On page 5 in line 2 “Figure 5”-> “Figure 2”

Response 4: Sorry for our carelessness. FTIR spectra is Figure 3 and not Figure 5. We have revised accordingly.

Reviewer 3 Report

In the present manuscript, the authors have studied the production of cationic lignocellulose nanofibers from agricultural waste and use them as adsorbents for the removal of dissolved and colloidal substances. 

This study does not have much to say, it does not offer any new aspects, it is completely repetitive and it has not even done an acceptable amount of work.

The Abstract part does not cover the most important findings.

The Introduction is too long. The flow of the introduction is not clear and it is not attractive to the readers. The authors must clearly explain the novel explorations made by them.

The discussion of the paper is weak and little scientific discussion has been presented to explain the obtained results. The authors should provide a conceptual focus on the interpretations and discussions and discuss results with some previously published studies.

The authors should made a better conclusion.

Author Response

Comments:

In the present manuscript, the authors have studied the production of cationic lignocellulose nanofibers from agricultural waste and use them as adsorbents for the removal of dissolved and colloidal substances.

This study does not have much to say, it does not offer any new aspects, it is completely repetitive and it has not even done an acceptable amount of work.

Point 1: The Abstract part does not cover the most important findings.

Response 1: The Abstract part has been modified accordingly.

Point 2: The Introduction is too long. The flow of the introduction is not clear and it is not attractive to the readers. The authors must clearly explain the novel explorations made by them.

Response 2: The Introduction has been shortened accordingly and we have made some changes in the Introduction to present our research more clearly to the reader.

Point 3: The discussion of the paper is weak and little scientific discussion has been presented to explain the obtained results. The authors should provide a conceptual focus on the interpretations and discussions and discuss results with some previously published studies.

Response 3: We have added and revised some of the Results and Discussion to enhance the explanatory and persuasive power of the article. We have added a conceptual focus on the interpretations and discussions and discuss results with some previously published studies so that readers can gain a more complete understanding of our research.

Point 4: The authors should made a better conclusion.

Response 4: The conclusion part has been modified accordingly.

Round 2

Reviewer 3 Report

The authors corrected manuscript according to reviewers suggestions.